# From Cultural Heritage Preservation to Art Craft Education: A Study on Taiwan Traditional Lacquerware Art Preservation and Training

**Chi-Sen Hung [1],*** , **Tien-Li Chen [2]** and **Yun-Chi Lee [3]**

[1] Department of Communication Design, National Taichung University of Science and Technology, Taichung 404, Taiwan

[2] Department of Industrial Design, College of Design, National Taipei University of Technology, Taipei 106, Taiwan; chentl@mail.ntut.edu.tw

[3] Doctoral Program in Design, College of Design, National Taipei University of Technology, Taipei 106, Taiwan; yunchilee0604@gmail.com

\* Correspondence: s6317666@nutc.edu.tw; Tel.: +886-913-227-832

**Abstract:** In Taiwan, preservation and training policies of intangible cultural assets are highly valued by the government. In this study, lacquerware art craft education as intangible cultural heritage is the subject of this study. We conducted in-depth interviews and secondary data collection to obtain research data and carried out a grounded theory data analysis method through expert meetings to explore the passing on education strategy of "lacquerware art craft" in Taiwan. Firstly, based on Bloom's educational objectives, the study analyzed three aspects of lacquer art education: cognitive, affection and skill, and proposed a "Lacquerware Art Passing-On Education Framework Diagram". Later, the analysis results of the grounded theory enable us to summarize the "Lacquerware art value and learning structure diagram". In this structure, it reveals that the Lacquerware artist's way of thinking about the craft levels can echo the system of the Three Extremes of the Tao in the Book of Changes and divide the value levels of creation into the levels of tools of livelihood, way of living and philosophy of life.

**Keywords:** intangible cultural heritage; preserving and training project; lacquer art; Taiwan





## 1. Foreword

The actual measures of passing on and preserving "cultural heritage" indicates how a nation-state respects ethnological culture and shows its maturity [1,2]. Taiwan's cultural history has several centuries of diversity and cultural memory resulting from different historical contexts of different ruling regimes and eras, and these historical contexts have set the tone of a cultural diversified Taiwan today. In 1995, the Council for Cultural Affairs (CCA) launched "The Plan for the Preservation and Transmission of Folk Arts" to preserve valuable intangible cultural assets in Taiwan. In 2005, with the promulgation of "Cultural Heritage Preservation Law", which clearly defined the mission of preserving traditional arts [3]; the preservation and transmission works were officially involved in general administrative affairs. In 2009, the local government and the central government began to register and designate the custodians of intangible cultural assets and implemented a preservation and transmission plan [4,5].

To understand the current development of Taiwan's intangible cultural assets inheritance and education, this study has explored relevant textual materials in the early stage and found that there are many published books and research reports that record Taiwan's lacquerware art that can clearly explain the historical context of development, record important events and figures, and introduce their works. However, we have discovered that the literature and these materials focus on the basic description of events; there is a lack of records on the promotion of inheritance education and educational concepts. Therefore,

this study chose to discuss the passing-on education strategy of "lacquerware art" with methods such as in-depth interviews, grounded theory, and other research methods. This study explores the cultural value of traditional lacquerware art in Taiwan, the philosophy, methods, and characteristics of lacquerware art education, and the art craft value structure. We intend to analyze the cultural values of traditional lacquerware art in Taiwan, the rationale of lacquerware education, and the methods and characteristics used by lacquerware conservators in education.

## 2. Development of Taiwan Lacquerware Art and Preservation and Transmission Education

### 2.1. Taiwan Lacquer Art and History of Origin

The lacquer tree is a deciduous tree that grows in East and South Asia countries, such as China, Japan, South Korea, Vietnam, Myanmar, Thailand, and India. Therefore, lacquerware art culture can be described as a type of art originating in the Asia. The lacquer tree takes eight years to mature, and the lacquer sap is obtained by cutting the bark of the mature lacquer tree at an oblique angle with a sharp blade and allowing the lacquer tree sap to naturally flow out. The lacquer sap is white and turns brownish red after solidification. After a period, it is oxidized and then transforms into a blackish dark brown color. Natural lacquer sap is an environmentally friendly material that beautifies, protects, and could be painted on various surfaces [6–9].

Before 1895, the lacquer tree cultivation technology and techniques for processing lacquer trees were unfamiliar in Taiwan. At that time, lacquer supplies relied on sea imports, so they were less popular and were only used for special occasions such as weddings and funerals to show the significance of the event. From 1895 to 1945, during the period of Japanese rule, the Japanese introduced lacquer trees from Vietnam to Taiwan in 1921. In 1916, the "Yamanaka Crafts Institute" was established, and in 1928, the "Taichung City Craft Education Center" was founded and began to cultivate Taiwanese lacquerware-making talents for lacquerware techniques. This was the earliest practice of lacquer craftsmanship in Taiwan, and Taiwan's lacquerware art culture originates from this time [10].

Taiwan's lacquerware art has only been developing for about a hundred years, a short period compared to China, which has an eight thousand years' history of developing lacquer craftsmanship. However, due to Taiwan's warm and humid climate, it provides a fertile environment for a number of natural materials, such as bamboo, lacquer, wood, etc. The promotion of lacquer tree cultivation and the training of lacquer professionals that began during the Japanese rule period enabled Taiwan to gradually break away from its dependence on lacquer craftsmanship and lacquer imports during the Ming and Qing dynasties, and become able to export its products to Japan and other countries [11].

Taiwanese lacquerware art originated in 1916 when the lacquerware artist Yamanaka (1884–1945) established the "Yamanaka Craft Institute" to start manufacturing and teaching classes and to serve as a base for product sales. In 1928, the Taichung Municipal Government and the Governor-General's Office jointly funded and subsidized the establishment of the "Taichung City Craft Education Center" and officially launched a systematic school, with the goal of passing on knowledge and skills related to craftsmanship. It established a woodworking sector and a paintwork sector and was divided into a two-year undergraduate major and a one-year research department. Ten students were recruited annually, and the center provided them with meals and daily necessities. Two years after graduation, it was stipulated that the graduated students must live in Taichung city, and they were obliged to do the relevant work. In 1936, the Taichung City Craft Education Center was changed into a private operation and was renamed "Private Taichung Craft Education School". In 1937, it was renamed "Private Taichung Polytechnics School" and expanded its enrollment from the original 10 pupils to 30 pupils. Applicants needed to meet the requirements of graduating from a small public school to improve the quality of enrolled students. There were lacquer and furniture subjects in the school, with a three-year period of study, and in that year the school initiated the admissions for "specialized students" with a two-year study period. After the National Government came to Taiwan in 1945, due

to the poor management of the school, the Taiwan People's Association bought the school and transformed it into the "Chien Kuo Craft Vocational School". In 1947, due to political factors, the Education Department of Taiwan Province announced that the school would be disbanded and abolished, ending the development of lacquerware art in the Japanese rule era [5,12].

### 2.2. The Development of Art Education in Taiwan

The earliest promotion of art education in Taiwan can be traced back to 1897 (Meiji 30) when the Taiwan Government-General's Office proposed a "Specific Plan for the Establishment of Public Schools in Taiwan" to initiate art education programs in Taiwan's education. In 1902 (Meiji 35), the "National Language School" added "Calligraphy and Painting" and began art education in Taiwan. Later, in 1934, the Japanese government established the Committee of Art Education in the Ministry of Education, which led Taiwan to the enlightenment period of art education. After the national government came to Taiwan, although curriculum standards were activated, local teaching materials were rarely included due to the overall political, economic, and social atmosphere. In 1982 and 1984, in accordance with the Implementation Rules for the Cultural Heritage Preservation Law, the Ministry of Education included the curriculum guidelines in local textbooks and began to apply historical, cultural, and artistic values as the editorial guidelines for textbooks. In 1989, the Standards Revision group for middle and primary schools added the "local art" to the standard curriculum, which was an important milestone in cultural education. From 1993 to 1995, the curriculum standards were drastically revised, and "Taiwan's local art textbooks" were included in the curricula of primary and secondary schools, until the local art education gradually entered normalization. In 1998, the Ministry of Education set the curriculum goals of art subjects to "Art and Humanities", which includes traditional opera, music, dance, fine arts, and indigenous art. Taiwan's cultural education tends to be diverse and has built the Taiwanese people's knowledge and recognition of local culture [12].

Due to the close cultural relationship between Taiwan and Japan, after the restoration of Taiwan, the people still used lacquerware, while in Japan, due to the rapid economic development and high labor costs, they began to seek ways to export related lacquerware supplies from Taiwan to Japan. The lacquerware art craftspeople cultivated during the Japanese Rule period, such as Huo-Qing Chen, Qing-Shuang Wang, and Gao-Shan Lai, were successively put into the production of lacquerware to provide the society and export needs at that time. The years between 1976 and 1986 were the heyday of lacquerware in Taiwan. There were many types of lacquerware products, such as wedding supplies, religious celebration supplies, tea trays, household altar tables, jewelry boxes, and flower utensils. At that time, there were about 40 lacquerware production factories. After the 1980s, with the rising labor costs in Taiwan and the forest protection policy, the traditional industries that required heavy manual processing gradually moved out. Since then, the domestic and foreign markets for lacquerware have gradually shrunk [13]

### 2.3. Policies and Programs for Craft Heritage Education in Taiwan

In recent years, Taiwan's craft inheritance education has been divided into formal school education, skill transfer programs held by the government in accordance with the cultural heritage laws, self-training courses from social groups, or courses handled by individual craftsmen. The development history of the identification and support of important technologies and persons for the preservation of cultural heritage is seen in early law records. Starting from the "Cultural Heritage Preservation Law" in 1982 and the "Implementation Rules on Cultural Heritage Preservation Law" in February 1984, intangible cultural assets such as art, folk customs, and related cultural relics were included in two chapters of the Japanese Cultural Property Protection Law [14]. In 1985, the Ministry of Education held the "Folk Art Heritage Awards" [15]. In 1989, it passed the selection and inheritance of education points of "Important Folk Art Artists", and the first group of national important folk art artists were selected, and the government began to promote

the education of the inheritance of various intangible cultural assets and skills. In 1995, the Council for Cultural Affairs implemented the "The Plan for the Preservation and Transmission of Folk Arts", which implemented four types of programs: preservation, training, research, and others. The training categories were further divided into traditional drama, music, and crafts. The plan was the most comprehensive in the categories, and projects supported by the government at that time. It was promoted in two phases, totaling more than eight years, and more than 2000 students received training reference numbers [16].

In 2005, the "Cultural Heritage Preservation Law" was revised, which merged the fourth chapter, "Folk Art" and the fifth chapter, "Folk Customs and Related Cultural Relics". They have put the old law as the content of the fifth chapter of the newly revised law, "Traditional Art, Folk Customs and Related Cultural Relics". At the same time, a registration and designation system was established to delegate authority and responsibility to local governments. The local governments investigate and register intangible cultural assets in various places, while the central government selects important ones from the traditional arts registered by the local governments for review and designation as "Important Traditional Arts, Important folk Customs and Related Cultural Relics", and they designated the first batch of important traditional art conservators and preservation groups in 2009. The "Cultural Heritage Preservation Law", which was revised and promulgated in July 2016, consolidates various types of traditional arts, folk customs, and related cultural relics into chapter seven "Intangible Cultural Assets" [3], and handles the "The Transmission Project for Important Traditional Arts Preservationists and Preservation Groups" (hereinafter transmission and preservation project) in accordance with Articles 92 and 97 and Article 34 of the "Rules for the Implementation of the Cultural Heritage Preservation Law" [15]. This plan is open for conservators or preservation groups to submit their own implementation plans. The conservators (often called artists or national treasures) will choose three to four art apprentices or recommend potential preservation and transmission artists to the Cultural Heritage Bureau. During the implementation of the plan, the artists must live, work together, and co-create with the conservator. By adopting the inheritance education strategies such as "oral transmission" and "incidental teaching method", it may achieve the learning effect of the mentoring system. In accordance with "The Transmission Project for Important Traditional Arts Preservationists and Preservation Personnel Completion Assessment Principles", the art apprentices participating in the project must finally go through an on-site implementation and demonstration, on-site oral examination, data review, and other parts of the evaluation reviewing the artistic ability, skill content, learning experience, humanistic literacy, knowledge, etc.; the apprentices would have to pass an evaluation that marks the preservation and transmission of the traditional art apprentice.

## 3. Research Method

Taiwan has been promoting programs and activities related to the inheritance of intangible cultural assets for more than 30 years, and thousands of students have acquired traditional skills as a result. Through the "Transmission Project for Important Traditional Arts Preservationists and Preservation Groups" promoted in recent years and the related data and the project results provided by the Cultural Heritage Bureau, these data have shown that in the year 2020, there have been 68 artists who have completed the workshop, including three artists in lacquerware craftsmanship. To thoroughly analyze how the preservation of lacquerware art craftsmanship carries out inheritance education and sorts out the context of education, this study chose to adopt an in-depth interview method, text data collection, and analysis, etc. The research processes planned in this study is shown in Figure 1. In Figure 1, it is explained that after confirming the purpose of the study, the in-depth interviews and secondary data collection were carried out to collect relevant data, and expert meetings were used to conduct a grounded theory, from which the "Lacquerware Art Passing-On Education Framework Diagram" and "Lacquerware

Art Value and Learning Structure Diagram" were proposed, and finally, conclusions and suggestions were made.

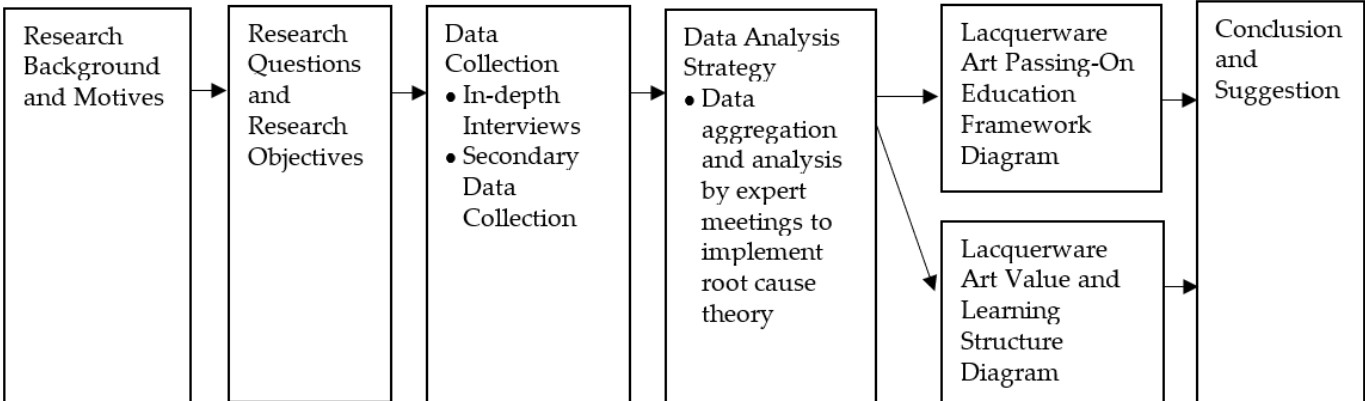

**Figure 1.** Research Processes.

*3.1. In-Depth Interview*

3.1.1. Interviewees

In this study, a total of nine experts, including two central government-recognized lacquerware craft cultural heritage conservators, one lacquerware craft and one cultural heritage conservator (recognized by Nantou County government), one graduate artist participating in the transmission and preservation project, four artists engaged in lacquerware art craft, and two public sector personnel who promoted the preservation and transmission project, were interviewed. The interview period was between November 2020 and March 2021. Table 1 is the Respondent's Information of this research.

**Table 1.** Respondent's Information.

| Respondent | Years of Experience | Age | Gender |
| --- | --- | --- | --- |
| Preserver of cultural heritage of lacquer art designated by the Ministry of Culture | 77 years | 100 years old | Male |
| Preserver of cultural heritage of lacquer art designated by the Ministry of Culture | 66 years | 85 years old | Male |
| The transmission and preservation project graduated art student | 47 years | 71 years old | Male |
| Lacquer artist | 25 years | 59 years old | Male |
| Lacquer artist | 28 years | 57 years old | Male |
| Lacquer artist | 25 years | 57 years old | Female |
| Lacquer artist | 5 years | 27 years old | Male |
| Cultural Assets Bureau/Deputy Secretary | 27 years | 59 years old | Male |
| Cultural Assets Bureau, Heritage and Folklore Section/Chief | 17 years | 48 years old | Male |
| National Taiwan Craft Research and Development Institute/Lacquer Art Researcher | 15 years | 44 years old | Female |

3.1.2. Theoretical Basis of Interview Content

The 12-year national basic education policy implemented in Taiwan has emphasized the core value that addresses the "core competency" thinking, which includes three major educational goals, which are knowledge, cognition, and skills. Additionally, the policy emphasizes three performance orientations that learning results should possess. The three performance orientations were developed by Bloom, Krathwohl, and Simpson et al., who divided teaching goals into three fields: cognitive, affective, and sensory [17–21]. With the abovementioned theoretical basis, the interview outline of this study was divided correspondently in three aspects (cognition, affection, and skills) to analyze the ideology of educators, learners, and policy agents on lacquerware art education strategies.

*3.2. Text Data Collection*

In addition of collecting first-hand data through in-depth interviews, this research also collected relevant research, lacquer art-related books, the preservation and transmission plan implementation rules, policy basis, achievement reports, and journals; these data were collected for reference and supporting evidence during inductive analyzation.

*3.3. Inductive Analysis Strategy*

The analysis strategy of this study was divided into two parts. First of all, for the education strategy of lacquer art, it was based on the three types of educational objectives (cognitive field, affective, and skill) as proposed by Bloom et al.; and we proposed the "Educational Framework for the Transmission of Lacquer Arts". The second part was conducted in grounded theory with in-depth interviews and a verbatim manuscript, and textual data were jointly encoded and analyzed. The grounded theory was carried out in three stages: open coding, axial coding, and selective coding. The grounded theory methodology used in this study was not chosen to be conducted via software, but rather through expert meetings. The main concern during the interviewing processes were whether the artists expressed their ideas in an metaphorical and abstract way. In order to avoid misunderstandings of the meaning of sentences by using software, the data analysis process of this study was conducted by three coding scholars who carefully discussed various data in conference. The three scholars, with backgrounds in cultural heritage preservation research, cultural education, educational policy, and educational evaluation, discussed the data and proposed a "Lacquerware Art Value and Learning Structure", and discussed the characteristics of lacquer art in this framework.

**4. Data Analysis**

*4.1. Lacquerware Art Education Strategy*

This study was based on the educational goals of cognition, affection, and skills as the analytical structure. From the in-depth interviews and textual data, three coding scholars firstly proposed the "Lacquerware Art Passing-On Education Framework Diagram", as shown in Figure 2 below.

This framework shows the comprehensive structure of the education of lacquerware art craft. In terms of cognition, the interviewees emphasized that learners should have an understanding of art history and the appearance and trends of the art market in modern society. Such knowledge is conducive to the confirmation of the future creation and development position of art apprentices. In other words, during the learning process, art apprentices should start thinking about and defining their own position in the art market to think about the direction of their possible future development. This cognitive mindset could benefit the art apprentice by confirming future creation and the development of art career. Regarding the knowledge of lacquer art technology, the artists should be able to understand and master lacquer-related knowledge, processes, tools, and pigment characteristics.

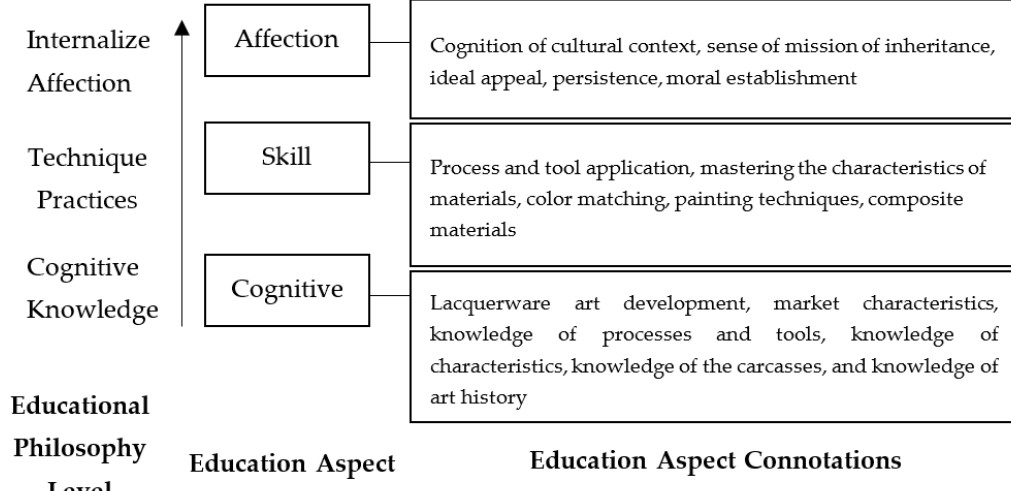

**Figure 2.** Lacquerware Art Passing-On Edu.

This framework shows the complete framework of education in lacquer art education. The knowledge of lacquer art should be understood and mastered in terms of lacquer-related knowledge, process, tools, and characteristics. As Taiwan's lacquer art heritage conservator Master Wang Qing-Shuang said, "Lacquer art can be used as the decorator of all kind of art craft, and it can be expressed on almost any material, such as ceramics, bamboo weaving, wood carving, and metalworking, so learners must learn various art types in a diversified way". For example, one of the interviewees, an artist who has been working in lacquer art for more than 30 years, specializes in the creation of ceramic-carcassed lacquerware. He chooses to use biscuit fired pottery as the medium for his lacquer art creation, and even experiments with a combination of glaze and lacquer materials, and shows the unexpected color expression under different temperatures! In other words, from the cognitive point of view, learners must master the properties of various materials, and through guidance and experience, they can gradually master the presentation of color.

In terms of skills, the transfer plan requires the art apprentices to study closely with the conservator during the period of transmission and preservation project and gradually master various lacquer techniques in their daily creation and work processes, especially the mastery of basic processes, tools, the characteristics and applications of composite materials, painting techniques, and style mastery. During the four-year period, although the transfer plan was conducted in the mode of "oral transmission and teaching" and "incidental teaching method", it started from the cultivation of basic aesthetics, such as sketching, observation, deconstruction, and color mastery; only then did it gradually enter a higher level of techniques, as well as the final stage of creation, innovative operation, and discourses. Master Wang Xian-Min, a Nantou County registered lacquer art cultural heritage conservator, said, "although there are only two important steps in the creation of lacquer art, one is basing and the other is decorative painting, it is extremely time-consuming and it is common that each piece takes 1–2 years to complete, which is an essential process to cultivate an artist's heart. As for the cultivation of aesthetic sense, we must start from sketching and drawing, and learn from nature by observing the colors of nature!" As for the techniques of lacquer art, they will be taught by means of apprenticeship, learning from imitation and copying. For example, an interviewed art apprentice who participated in the Transmission and Preservation project said: "We will first learn to draw peonies, which represent wealth and prosperity, or peacocks, which represent good luck and kindness, from our tutors. Though this has market considerations, it is essential to have these basic skills training in order to understand the application of color materials and the mastery of techniques".

In terms of cognition, the interviewees emphasized that learners should have a deeper understanding of the history of art and of the patterns and trends of the art market in

modern society. Such cognition is conducive to the confirmation of future creation and the development positioning of art apprentices, in other words, during the learning process, the art apprentices should also begin to think about and define their own position in the art market in order to determine the potential future development.

Finally, affection is regarded by the interviewees as the most central and important high-level educational goal in the learning process. If inheritance education can achieve the learning goal of affection, it signals a change in the personality, character, attitude, and values of the art apprentices. Judging from the connotation of its educational orientation, it includes understanding and recognition of the cultural context of the development of lacquer art craftsmanship, the sense of mission of inheritance education, the values, and the ability to comment on the formation of creative ideas, etc. In addition, the current "Transmission and Preservation Project" implemented by the Bureau of Cultural Heritage for allows preservationists or preservation groups to submit their own implementation plans, and the training period is three to four years, starting from the training of basic skills, and through continuous practices and creation, the students become capable of creating and interpreting artworks independently. The lacquer art conservator who participated in this project said: "The development of lacquer art in Taiwan is different from Japan's single-technology specialization, since the development of lacquer art in Taiwan is rather late, so the creation of lacquer art artists in Taiwan usually incorporates different techniques, resulting in a special style of lacquer art in Taiwan. The lacquer artists often develop unique ways of presentation, using materials such as mother-of-pearl, gold leaf, eggshell, gold powder, and paste to express different color textures in their surface decorations, infusing their own spiritual energy into their works".

In this study, interviews with conservators, apprentice artists, and lacquer craftsmen revealed that the spirit of the "Transmission and Preservation Project" is in line with the traditional folk art apprenticeship system Figure 2. presents the structure of the cultural assets preserved by the lacquer art of Taiwan in the educational transmission. In the current education of lacquer art in Taiwan, primary emphasis is placed upon the cultivation of techniques, such as lacquer materials, aesthetics, painting techniques, material knowledge, and the application of composite media, etc. In this process, the students are involved in the work, creation, and even living together with the conservators. During the process, apprentices not only learn lacquer techniques and gain experience from the conservators but also learn the conservators' attitude towards the quality of their works and the constant pursuit of perfection, so as to establish a good creative attitude for the apprentices.

In the process of knowledge cultivation, the conservators often require the apprentices to diversify their knowledge acquisition, not only in terms of knowledge of lacquer art culture, lifestyle, market characteristics, lacquer materials, lacquer techniques, and development history, but also in terms of learning and reading across different craft categories. After three to four years of training, the conservators begin to guide the apprentices towards the construction of their own creative concepts, gradually moving from the stages of "capable of making", "capable of thinking", and finally "capable of self-explanation", to cultivate the apprentices' emotions and identification with the art of lacquer craft.

During the analysis process, this study also discovered that the past learning experience of an intangible cultural heritage conservator profoundly affects the teaching methods of the conservator as an educator. Although the transmission and preservation project was based on the model of "oral transmission and teaching" and the "incidental teaching method", the system of educational content still requires the Cultural Heritage Bureau to arrange professional consultants to assist the conservators, which benefits the systematization of educational content. As a young generation of transmission and preservation project art apprentices, they also achieve the mission of organizing the experience and knowledge of the cultural heritage conservators throughout their lives. The interviewee, the Deputy Secretary of the Cultural Heritage Bureau, believes that intangible cultural assets are similar to an organic body in the context of cultural development, which grows and declines depending on the time and space. However, intangible cultural assets are

inherited by "people", and when a cultural asset preserver passes on his or her knowledge, skills, and feelings to the next generation, we should keep an open mind and let it develop naturally!

### 4.2. The Value Level and Learning Level of Lacquerware Art

In addition to the educational strategy of lacquer art in the previous paragraph, this study analyzes the in-depth interviews and textual data from the viewpoint of educational and artistic connotations during theoretical development. It is expected that through the expert meeting strategy, systematic procedures will be adopted to gradually disclose the viewpoints and meanings of the content of the materials, expecting to obtain a more psychological inclination, artistic discourse, and spiritual discourse of creation.

The result of the grounded theory enables us to develop a "Lacquerware Art Learning Framework", which corresponds to the spiritual structure of arts and crafts values, as well as the creative structure. Therefore, after the expert meeting, we proposed a "Lacquerware Art Value and Learning Structure Diagram" (Figure 3) by integrating the three parts, namely, the learning structure, the spiritual structure, and the creation structure. Due to the limited space, we are unable to fully present all the details of the integration process in this study, so we have placed a comprehensive and simplified Table A1 of the "Lacquer Learning Framework Code" in the Appendix A of this study.

From reflections taken at the expert meeting, we attempt to unveil the connotation of the artistic value of lacquer art and the learning levels of the trainees. This study concluded from the expert meeting that the lacquer artists' contemplation of the craft levels echoes the three extremes of the Tao system in the *Book of Changes* as well as the core of Lao Tzu's *Tao Te Ching*, which quoted "The Way gave birth to one. One gave birth to two. Two gave birth to three. Three gave birth to all things. All things carry yin and embrace yang. They reach harmony by blending with the vital breath". The "Lacquerware Art Value and Learning Structure Diagram" is summarized as shown in Figure 3 below:

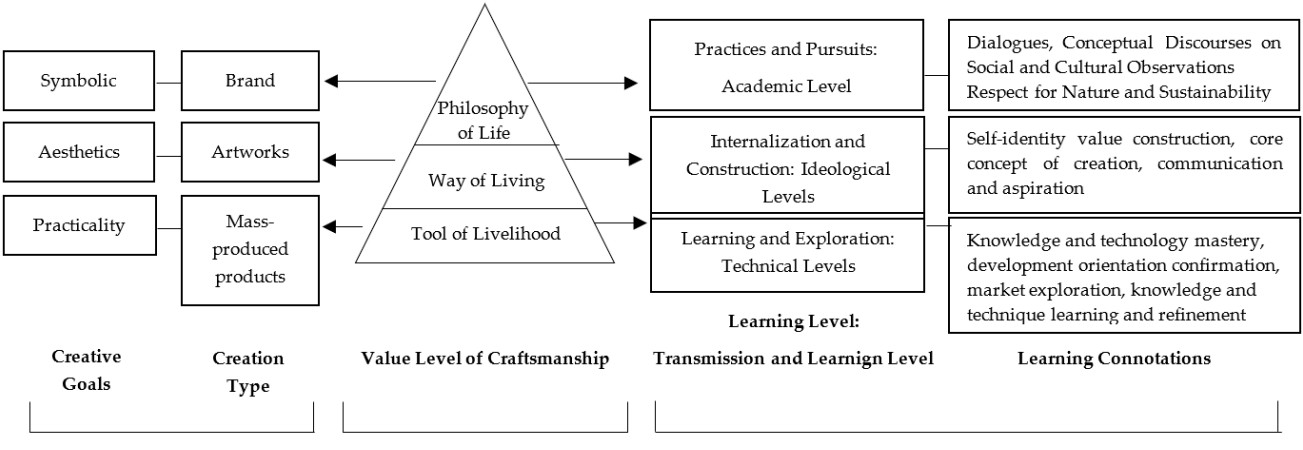

**Figure 3.** Lacquerware art value and learning structure diagram.

Lacquerware art preservation is not only the cultivation of techniques and knowledge but also the process of cultivating the personality and character of the art apprentices. The value level of creation can be divided into three levels: livelihood, living, and life.

The "Tool of Livelihood" corresponds to the level of learning, where the emphasis is on the cultivation of knowledge and skills of the art students. During the training, the interviewees, who are the preservers of lacquer art in Nantou County, said that in the process of learning lacquer art, they must rooted in their fundamental lacquer art skills and abilities so that they might be able to connect with the market and maintain their livelihoods after graduating. The "tool of livelihood" level is achieved in basic

education and learning, which focuses on the cultivation of the knowledge and skills of transmission and preservation project art apprentices. First of all, the basic ability of art apprentices to integrate into the market to make a living after graduation is the main focus. In terms of lacquerware art, it is necessary to be familiar with the process, master the characteristics of various lacquer liquids, flexibly applying various tools and composite materials, and practice on different carcasses. The lacquerware art decoration techniques include lacquer applying, delineation, embedding, Maki-e, the Raden technique (it is a technique in which pieces of shell are attached to the surface of the carcass and then grinded to produce the characteristic color of the shell, and it is called "Luo-Dian" in Chinese,) multiply lacquer layers, stacking, and etching carved lacquer, etc. The surface of lacquer crafts can be treated with delicate painting techniques revealing different colors through different procedures such as polishing, delustering, and consider the texture, natural texture, cracks, and other variants. There are more composite materials such as jade, gold leaf, mother of pearl, eggshells, etc. These embellishments and decorations are the reasons that make lacquerware art unique and varying in visual changes [22]. In addition, learning at this stage needs to cultivate the market demand and consumption characteristics of the domestic and international markets. From the interviews we learned that even the cultural heritage conservators continue to emphasize innovative expressions and visual symbols in each creation, which is also one of the critical points of inheritance education. Such an attitude of constant search for newness, change, and improvement is one of the key points of heritage education.

The "Way of Living", proposed in Figure 2, the Lacquerware Art Value and Learning Structure Diagram, discusses the internalization of knowledge and skills by creators, who can gradually construct their creative values and beliefs and communicate their ideas and demands with the market in the form of art collections. Artistic literacy stems from the combination of beauty and philosophy. It has gradually moved away from daily necessities and artifacts in the mass consumer market and towards art collections with conceptual expressions, echoing people's yearning for life attitudes and lifestyles. The interviewee, who is a lacquer artist, said: "Lacquer art itself is an extraordinary craftsmanship, and students must recognize this and be determined to continue creating. It is important not to go overboard with superficial, superficial, experiential learning for a short period of time, and that is why the 'transmission and preservation project' must be at least 3 years long".

Due to the creative process of the lacquer process, it is necessary to continue the repetitive process of painting, polishing, repainting, and re-polishing, which has become the process of cultivating temperament, internalizing emotions, and comprehending the truth. Therefore, the highest level of value demonstrated by lacquerware art craftsmanship is the "Way of Life".

In the "Philosophy of Life" level, creators have transcended the market mindset and emphasized self-practice instead and express their views and respect for society, culture, and ecology through their works. The interviewees who used basket as a form of lacquer art creation said, "our inspiration and materials are all from nature, and many crafts in Taiwan are are taken from nature and used in nature. Creation itself is symbiotic, coexisting and in dialogue with nature, such as bamboo art and lacquer art. The work will eventually decay and return to nature, just like each of our existence in the world. At such a level, art has transcend into beliefs that the art apprentices may carry on to preserve these intangible cultural heritage through their skills, and with such belief this will be the motives for future preservation, activation, innovation, and pass on as a lifelong task so that the skills can be passed on sustainably". [Figures 4 and 5 (photographed by author)].

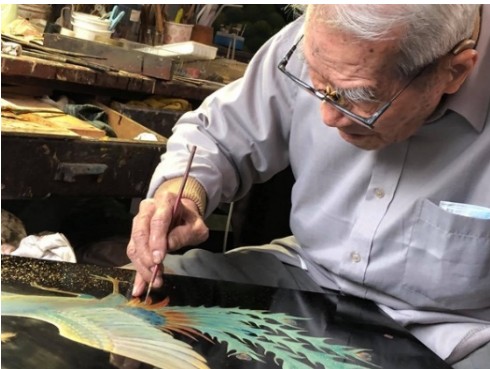

**Figure 4.** The lacquer art cultural heritage conservators interviewed for this study: Qing-Shuang Wang. The interviewee personally demonstrates the drawing process and technique.

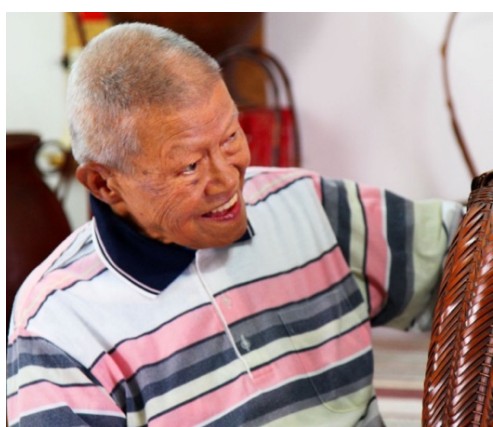

**Figure 5.** The lacquer art cultural heritage conservators interviewed for this study: Lee Rong-Lieh, the interviewee who combines bamboo art with lacquer art to create the art form of "basket lacquerware".

## 5. Conclusions and Suggestion

Due to the modern lifestyle, traditional cultural heritages are rapidly vanishing due to changes in market characteristics. Taiwan has been promoting projects related to the inheritance of intangible cultural assets for more than 30 years. This study focuses on the "Transmission Project for Important Traditional Arts Preservationists and Preservation Groups" implemented in recent years as a case study. The cultural heritage conservators, art apprentices, craftsmen, and public sector personnel who have promoted the project were invited to conduct in-depth interviews.

The study conducted a grounded theory analysis by means of expert meetings and proposed "Lacquerware Art Passing-on Education Framework Diagram" and "Lacquerware Art Value and Learning Structure Diagram" containing three sub-frameworks: "Lacquer Art Learning Sub-Framework", "Spiritual Sub-Framework", and "Creative Sub-Framework" in response to the research objectives and analyze the strategy of art craft education in Taiwan's lacquer craftsmanship inheritance education, as well as the value level and learning level of lacquer craftsmanship. However, there are many types of craftsmanship with different cultural contexts and social relations, processes, techniques and tools, and deep knowledge systems, which are topics worthy of future research.

**Author Contributions:** Conceptualization, C.-S.H. and T.-L.C.; Writing–original draft, C.-S.H.; Writing—review and editing, C.-S.H. and T.-L.C.; Supervision, T.-L.C.; Investigation, C.-S.H. and Y.-C.L.; Data curation, Y.-C.L.; Analysis, Y.-C.L. All authors have read and agreed to the published version of the manuscript.

**Funding:** This research received no external funding.

**Institutional Review Board Statement:** Not applicable.

**Informed Consent Statement:** Not applicable.

**Data Availability Statement:** Data sharing not applicable.

**Conflicts of Interest:** The authors declare no conflict of interest.

## Appendix A

**Table A1.** A synthesis of learning framework codes compiled from grounded theory (Simplified Table).

| Selective Coding | Axial Coding | Open Coding |
|---|---|---|
| Learning and Exploration: Technical Levels | Development Positioning Confirmation | • Positioning for the future based on aspects of the market<br>• Positioning style from the technical aspect and expression<br>• The choice of carrier for the presentation of lacquer art<br>• ..., etc. |
| | Exploring the market | • The Development of Lacquer Craft<br>• Marketing nature of lacquer art creation<br>• Characteristics of domestic and international markets<br>• ..., etc. |
| | Knowledge and Technique Learning and Refinement | • Understanding and production of lacquer materials, painting techniques, surface treatment techniques, composite materials<br>• The production and characteristics of the body<br>• Observation of nature<br>• ..., etc. |
| Internalization and Construction: Ideological Levels | Self-identity value building | • Lacquer art is an ultimate craftsmanship, and students must have this understanding and the determination to continue creating<br>• To learn the concepts and techniques of national conservator and to take on the mission of preserving culture<br>• ..., etc. |
| | Creation Core Concept | • The shape of the body, whether it is flat or three-dimensional, it already has the existence of a theme<br>• Lacquer art can be a decorator for different kinds of artworks.<br>• The creator must pour his spirit into the work<br>• ...etc. |
| | Communication and Aspiration | • Artwork needs to find a market, whether it is an artistic market or a commercial market<br>• The traditional market has traditional cultural patterns that are accepted, and after modernization, new patterns and new meanings emerge.<br>• In the face of the changes in modern life, how can artworks speak to the modern public?<br>• ...etc. |

**Table A1.** *Cont.*

| Selective Coding | Axial Coding | Open Coding |
|---|---|---|
| Practice and Pursuit: Academic Level | Philosophy | • At the end of the day, the art apprentice may also become the next preservationist, and at this time, he or she must develop his or her own set of discourses.<br>• Such cultural skills must be passed on to the next generation. Of course, at this time, you will have your own teachings, and you will also tell the next generation about your insistence on the art.<br>• ...etc. |
| | Respect for Nature and Sustainability | • Before painting, you must sketch, which is a kind of atmosphere given to us by nature, and it also tells us that everything comes from nature.<br>• Lacquer art is made from the most natural materials, and many crafts in Taiwan are like this. It is taken from nature and used in nature, just like the creation, it is also a symbiosis, coexistence and dialogue with nature, and the moment the work disappears, it is also a return to nature, just like we all exist in the world.<br>• ...etc. |
| | A Dialogue of Social and Cultural Observations | • Only when there are people can there be art. In other words, art is appreciated by people, art is interpreted by people, and people are moved by art, and artists are in dialogue with people in this way.<br>• Honestly speaking, the craftsmanship itself is similar, like the high Maki-e painting in lacquer art, which actually has the same concept as the three-dimensional sculpture and carving, especially the aesthetic sense itself is similar. You wouldn't say that the way I create lacquerware is not related to the way I think about sculpture, would you?<br>• I think if a craft is not able to sustain its livelihood, then it will be eliminated sooner or later. The "Transmission and Preservation Project" has contributed to the education of the heritage, but it still has to return to the creator himself, how to take this craft and create a dialogue with the modern society and culture.<br>• ...etc. |

Note: Due to the limitation of space in this article, the contents of this table have been simplified and presented in the form of extracts of highlights.

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
