# Peer review of "From Cultural Heritage Preservation to Art Craft Education: A Study on Taiwan Traditional Lacquerware Art Preservation and Training"

_education, doi:10.3390/educsci11120801_

Round 1
Reviewer 1 Report
1. In this article, the author has interviewed 1 graduate artist, 4 artists, and 3 public sector personnel, who are related to lacquer art. However, the analysis did not provide a detailed analysis of Taiwan traditional lacquerware art preservation and training, nor did it distinguish between different informants. What is the difference between the interviewees on the preservation and training of lacquer art?
2. Although the author claims that this article adopts grounded theory research methods, it does not show any content of open coding, nor does it establish the main categories and sub-categories, and does not quote any important words of the interviewee as evidence in the writing.
3. The title of this article is From cultural heritage preservation to color education, but in the analysis of the fourth section, color education in lacquer art is hardly involved. In other words, the content shown is not consistent with the main title.
4. Basically agree with the three levels of content shown in figure 2 lacquerware art value and learning structure diagram, but this concept is not innovative in the craft research and can be used in almost all craft creations. Therefore, it is recommended to develop a unique value and learning structure diagram for lacquer art.
Author Response
- Thank you for your suggestions. In the revised version, we have enhanced the perspectives of respondents from different backgrounds in this study in order to describe their views on the preservation and training of lacquer art from different perspectives.
- Thank you for your valuable suggestions. In the original version, we indeed oversimplified the process and exposition of the grounded theory coding, although in the original Figure 2 we tried to show and explain the results after axial coding, but this seems almost too simplified.
We are grateful for the important reminders given by the reviewers. In the revised version, we have enhanced the discourse on the coding process of the grounded theory, supplemented some of the coding information of the grounded theory in the appendix, and quoted some of the significant quotes from the interviewers in the discussion of the study as supporting evidence.
- We appreciated the valuable suggestions of the members, and in the revised version, we have once again highlighted the characteristics of lacquer art color in the section 4.1. In fact, in the process of analyzing the data, we found more information about the artists' comprehensive painting techniques, materials, creative concepts, and even artistic persistence. These information are truly fascinating, and we would like to share this exclusive Taiwanese artistic spirit to the world through our study and we have organized these information in the section 4.2.
We thank the reviewers for this important reminder, so in the revised version we have reorganized the correspondence throughout the paper to try to make our key messages clearer.
- We appreciate the valuable suggestions of the reviewers, and indeed, as the reviewers said, the level of art craft values as in Figure 2 appears to show the levels of various types of crafts. However, this result is the outcome of an expert meeting with three university professors from the backgrounds of cultural heritage preservation research, cultural color education, and educational policy and evaluation, in which we analyzed and discussed the data systematically in a theoretically grounded manner.
To this end, we again discussed with three university professors to review whether such a framework is exclusive to the value level of lacquer art. As a result of this discussion, the experts concluded that the characteristics of lacquer art are embedded in the "Learning and Exploration: Technical Levels" part of this framework, which is more technical in nature. With the pursuit of higher value in art, artists no longer seek for techniques or appearances, but for creative concepts and spiritual expressions. At such a level, it is beyond the category of craftsmanship, hence, the result of such a structure.
We are once again grateful to the reviewers for their suggestions, which gave us the opportunity to rethink the rationale and depth of the discussion, and in the revised version, we have again strengthened the discussion about this part.
Reviewer 2 Report
The article has a clear structure and interesting results that apply to lacquer arts in Taiwan as well as, more broadly, to ideas about preservation of culture and enrichment and engagement for individuals through art.
Moderate editing is required, mostly for style.
8-9 change Cognitive, Affective etc. to lowercase
10 lacquer art preserver--awkward. Maybe "conservators of lacquer art and cultural hertitage and government apprentices to preservation and training projects.
15-delete "majorly" (awkward and unnecessary)
17 change interrelationships to "relationships"
20 "affection" --explain this or at least characterize with a couple of words: "affection of..."
26 Taiwan cultural history has several centuries of diversity and cultural memory resulting from...
30 valuable intangible cultural assets in Taiwan
42 "surface of the event" is unclear--what does that mean?
43 Unless "inheritance education strategy" is a widely used and understood term, I would say, "this study chose to use what is called 'inheritance education study' to..."
46 explain "the education philosophy" (used in Taiwan?) "the education model" (again, whose education model?)
50-51 Southeast Asia does not include China, Korea, or Japan (these are in East Asia) or India (South Asia).
52 Use "Asia" instead of "the East" (outdated term)
61-62 to show nobility and respect of the event (awkward-sounding)
67 delete "initiates since then"; substitute "originates from this time.
68-70 delete some words and switch around to: "...has only developed for about a hundred years and is short compared to China, which has an eight thousand year history of developing lacquer craftsmanship."
70 swap "growing" for "fertile"
72 "talent cultivation" is awkward. Maybe craftsmanship, ingenuity, and technique.
83-84 Fragment sentence. Needs a subject (the center provided them with meals...)
85 were (instead of was)
88 "and expanded its enrollment..."
90 delete "or above"
98-99 check your dates or clarify with an explanation. How does it date to 1902 and then was established in 1934?
101 substitute "became" for "was"
106 guidelines (add an 's')
111 schools, until the local art education
\113-change "making the goals" to "making them"
113 Only need to use "traditional" as a modifier to the first noun, not for every one "traditional opera, music, dance, fine arts..."
115 "tends to be diverse (no comma) and has built the Taiwanese..."
117 people still used (not "are still accustomed to use")
119 "they began to seek ways to export lacquerware supplies from Taiwan..."
120 by "art talents" do you mean craftspeople or experts? Or lacquerware making skills? "Art talents" doesn't quite make sense in English.
122-23 "at that time. The years between 19786 and 1986 (no comma) were the heyday"
128-29 "for lacquerware gradually shrank."
133-34: "About the development..." This is a fragment sentence--needs a subject and verb
136 "1984, (no 'the') tangible cultural assets, such as art, folk customs, and related cultural relics, were included in two chapters of the Japanese Cultural Property Protection Law."
145 Use "traditional" only with the first noun
146-47 Seems circular: The plan was covered by the government-promoted plan?
154 delete "will"; just say "local governments investigate and register..."
157 capitalize Folk, break up sentence: "...'Folk Customs and Related Cultural Relics.' They designated..."
165: change "or" to "often" (often called)
165- spell out all numbers under 10: three to four art apprentices. (delete "by themselves"
168-170 Fragment sentence (reads more like a heading, like 133-34)
172 "transmission of art"
186 "craftsmanship carries out"
187 "adopt an in-depth"
192 "one central government" (spell out number; no capitals for central gov)
193 "Taichung city government"
199 "in Taiwan has emphasized"
234 What does "confirm development positioning" mean? Clarify by rewording or explaining.
240 "especially the mastery of basic processes, tools..."
241 "During the four-year period, although the transfer plan..." (by this point the reader isn't sure what "it" refers to, so restate here)
248 "affection, it signals a change in the personality..."
253 "In this study, the results of..."
254 "etc., showed that the spirit..." (Apart from some changes needed for language, conceptually this part of the paper is very interesting!)
276 needs a source ("error! Reference not found")
281-300 This part of the paper opens up new avenues for thinking about the conclusions and how art preservation and conservation, along with cultivating traditional craft skills, makes meaningful differences in culture and people's lives. Great!
296 Raden? Do you mean rattan?
297-98 Sounds like a heading--reword into a full sentence.
299 "From the interviews we learned..."
303 "The Way of Living"--is this a text? If it is a book, italicize it and give the author and cite it. If you are referring to it as a benchmark, then introduce and explain it on the first mention.
319 Error! Reference source not found.
Author Response
The article has a clear structure and interesting results that apply to lacquer arts in Taiwan as well as, more broadly, to ideas about preservation of culture and enrichment and engagement for individuals through art.
We appreciate the reviewer's positive remarks, and we share the same view that Taiwan's culture must put in the effort to be seen by the world. As for the transmission of intangible cultural heritage, there are too many efforts to be recorded, both from the government and from the private sector, and to share this experience with the world!
Moderate editing is required, mostly for style.
8-9 change Cognitive, Affective etc. to lowercase
Corrections were completed in accordance with the reviewer's comments.
10 lacquer art preserver--awkward. Maybe "conservators of lacquer art and cultural hertitage and government apprentices to preservation and training projects. 10
We thank you for your great suggestions, and we have completed the corrections based on the comments.
15-delete "majorly" (awkward and unnecessary)
Corrections were completed based on reviewer comments.
17 change interrelationships to "relationships"
Corrections were completed based on reviewer comments.
20 "affection" --explain this or at least characterize with a couple of words: "affection of..."
In response to reviewer comments, this has been revised to the affection of artistry and craftsmanship.
26 Taiwan cultural history has several centuries of diversity and cultural memory resulting from...
We are grateful to the reviewer for your careful and valuable suggestions, and we have completed the corrections based on the suggestions.
30 valuable intangible cultural assets in Taiwan
Corrections were completed based on reviewer comments.
42 "surface of the event" is unclear--what does that mean?
We are sorry for the misleading terminology used in the original version. We have made a correction in the revised version.
43 Unless "inheritance education strategy" is a widely used and understood term, I would say, "this study chose to use what is called 'inheritance education study' to...
We did miss such details, thank you for the careful and valuable comments of the reviewers, and corrections were completed based on the comments.
46 explain "the education philosophy" (used in Taiwan?) "the education model" (again, whose education model?)
We would like to thank the reviewer for providing valuable suggestions, which we have supplemented explanation that should make the purpose of this study better understood by the reader.
50-51 Southeast Asia does not include China, Korea, or Japan (these are in East Asia) or India (South Asia).
Corrections were completed based on reviewer comments.
52 Use "Asia" instead of "the East" (outdated term)
Corrections were completed based on reviewer comments.
61-62 to show nobility and respect of the event (awkward-sounding)
Many thanks to the reviewer for this suggestion, we have made a few refinements here and have completed the correction.
67 delete "initiates since then"; substitute "originates from this time.
Thank you for providing invaluable advice and we have completed the corrections based on your suggestions.
68-70 delete some words and switch around to: "...has only developed for about a hundred years and is short compared to China, which has an eight thousand year history of developing lacquer craftsmanship."
Corrections were completed based on reviewer comments.
70 swap "growing" for "fertile"
Corrections were completed based on reviewer comments.
72 "talent cultivation" is awkward. Maybe craftsmanship, ingenuity, and technique.
Thanks to the valuable suggestions, we have modified it here to "craftsmanship".
83-84 Fragment sentence. Needs a subject (the center provided them with meals...)
We thank the reviewer for your valuable and careful suggestions and have completed the corrections based on the reviewer's suggestions.
85 were (instead of was)
We thank the reviewer for your valuable and careful suggestions and have completed the corrections based on the reviewer's suggestions.
88 "and expanded its enrollment..."
Corrections were completed based on reviewer comments.
90 delete "or above"
Corrections were completed based on reviewer comments.
98-99 check your dates or clarify with an explanation. How does it date to 1902 and then was established in 1934?
Thank you for the suggestion, it made us re-examine the correctness of the original version of the writing,In the new version, we have made adjustments and data updates, and have completed relevant corrections.
101 substitute "became" for "was"
It has been corrected, thanks to the reviewer's suggestion.
106 guidelines (add an 's')
Corrections were completed based on reviewer comments.
111 schools, until the local art education
Corrections were completed based on reviewer comments.
\113-change "making the goals" to "making them"
Corrections were completed based on reviewer comments.
113 Only need to use "traditional" as a modifier to the first noun, not for every one "traditional opera, music, dance, fine arts..."
Corrections were completed based on reviewer comments.
115 "tends to be diverse (no comma) and has built the Taiwanese..."
Corrections were completed based on reviewer comments.
117 people still used (not "are still accustomed to use")
Corrections were completed based on reviewer comments.
119 "they began to seek ways to export lacquerware supplies from Taiwan..."
Corrections were completed based on reviewer comments.
120 by "art talents" do you mean craftspeople or experts? Or lacquerware making skills? "Art talents" doesn't quite make sense in English.
We thank the reviewer for suggesting that we modify it to "craftspeople" here.
122-23 "at that time. The years between 1976 and 1986 (no comma) were the heyday"
Corrections were completed based on reviewer comments.
128-29 "for lacquerware gradually shrank."
Corrections were completed based on reviewer comments.
133-34: "About the development..." This is a fragment sentence--needs a subject and verb
We thank the reviewer for your valuable and careful suggestions and have completed the corrections based on the reviewer's suggestions.
136 "1984, (no 'the') tangible cultural assets, such as art, folk customs, and related cultural relics, were included in two chapters of the Japanese Cultural Property Protection Law."
Corrections were completed based on reviewer comments.
145 Use "traditional" only with the first noun
Corrections were completed based on reviewer comments.
146-47 Seems circular: The plan was covered by the government-promoted plan?
Thanks to the reviewer for the reminder, we have modified the sentences.
154 delete "will"; just say "local governments investigate and register...
Corrections were completed based on reviewer comments.
157 capitalize Folk, break up sentence: "...'Folk Customs and Related Cultural Relics.' They designated..."
Corrections were completed based on reviewer comments.
165: change "or" to "often" (often called)
Corrections were completed based on reviewer comments.
165- spell out all numbers under 10: three to four art apprentices. (delete "by themselves"
Corrections were completed based on reviewer comments.
168-170 Fragment sentence (reads more like a heading, like 133-34)
Corrections were completed based on reviewer comments.
172 "transmission of art"
Corrections were completed based on reviewer comments.
186 "craftsmanship carries out"
Corrections were completed based on reviewer comments.
187 "adopt an in-depth"
Corrections were completed based on reviewer comments.
192 "one central government" (spell out number; no capitals for central gov)
Corrections were completed based on reviewer comments.
193 "Taichung city government"
We reconfirm the identity of the interviewee who is registered as a lacquer art conservator by the Nantou County Government, and we have corrected it in the revised version.
199 "in Taiwan has emphasized"
Corrections were completed based on reviewer comments.
234 What does "confirm development positioning" mean? Clarify by rewording or explaining.
What we meant here is that the art teacher should not only be teaching technical skills, but also guiding the students to find their own position and development in the art market independently after they have finished their studies.
We have modified the description, thanks to the reviewer for the reminder.
240 "especially the mastery of basic processes, tools..."
Corrections were completed based on reviewer comments.
241 "During the four-year period, although the transfer plan..." (by this point the reader isn't sure what "it" refers to, so restate here)
We thank the reviewer for the careful reminder. We have completed corrections based on the review comments in the revised version.
248 "affection, it signals a change in the personality..."
Corrections were completed based on reviewer comments.
253 "In this study, the results of..."
Corrections were completed based on reviewer comments.
254 "etc., showed that the spirit..." (Apart from some changes needed for language, conceptually this part of the paper is very interesting!)
In this review, other reviewers suggested that we could add some key phrases of the respondents and the development process of the grounded theory.
Once again, thank you very much for giving us valuable advice.
276 needs a source ("error! Reference not found")
We apologize for the error in the archiving process in the original version, which has been fixed in the updated version.
281-300 This part of the paper opens up new avenues for thinking about the conclusions and how art preservation and conservation, along with cultivating traditional craft skills, makes meaningful differences in culture and people's lives. Great!
We thank you for the support and recognition and hope that such a study will demonstrate to the world Taiwan's efforts in the transmission of crafts and education. In this revised version, we have provided more information about the interviews, which we expect will make the whole study more in-depth and valuable!
296 Raden? Do you mean rattan?
Raden here refers to a technique in which the artist uses thin sheets of shells to adhere to the surface of the lacquer art. In Taiwan, the name is translated from Japanese as Ra-den, which is a technique used in the creation of lacquer art.
We wonder if it would be a good idea to add the Chinese character "螺鈿" to the revised version. If the reviewer doesn't think it's appropriate, please give us another suggestion and we'll remove it.
We would appreciate if the reviewers could give us their valuable suggestions, thank you!
297-98 Sounds like a heading--reword into a full sentence.
Corrections were completed based on reviewer comments.
299 "From the interviews we learned..."
Corrections were completed based on reviewer comments.
303 "The Way of Living"--is this a text? If it is a book, italicize it and give the author and cite it. If you are referring to it as a benchmark, then introduce and explain it on the first mention.
In Figure 3 (newly revised version), the "hierarchy of values of crafts" proposed by the expert meeting is expressed, in which the "hierarchy of values of crafts" can be divided into three levels from the bottom to the top, namely, "the tools of livelihood", "the way of living", and "the way of life", and such a spiritual structure can be corresponded to the three levels of the lacquer art learning structure and the three levels of the creation structure.
We thank the reviewer for this suggestion. We have further elaborated on these three levels in the updated version.
319 Error! Reference source not found.
We apologize for the error in the saving process in the original version, which has been fixed in the revised version.
Reviewer 3 Report
There are some observations related with structure and results analyses. the text needs to be reviewed. Check the attached file.

Author Response
Many thanks to the reviewer for giving us such valuable suggestions, which have enabled us to thoroughly rethink, reorganize, and propose an improved version of the study. We look forward to such amendments that will meet the reviewers' requirements. Thank you from the bottom of our hearts! 1.Thanks to the reviewer's valuable suggestions, we have supplemented the flow chart for this study in the updated version. 2.Thanks to the valuable comments, we have divided the content of section 2 into three subsections and have completed the corrections in the new version. 3.Thanks to the reviewers' suggestions, in the updated version we have divided the content of Chapter 2 into three subsections, simplified some sentences, and added corrections to other reviewers' comments, which should make it more readable. Thank you from the bottom of my heart for your valuable suggestions! 3.1.2 Based on the reviewers' comments, we have supplemented the updated version with the basic information of the respondents in Table 1. Thank you for your careful reminder, we have made the corrections in the updated version in accordance with your comment. 4. We apologize for the error in the archiving process in the original version, which has been fixed in the updated version. 5.We thank you for the valuable suggestions, and in section 5, Conclusion and Suggestion, we have reorganized the phrases and strengthened the relationship between the purpose of this study and the results.Round 2
Reviewer 1 Report
- Although the title of the thesis mentions color education, in fact, the inheritance of lacquer art does not only teach color, but includes materials, techniques, creation and other aspects. Therefore, it is recommended that the color education in the title can be modified to craft education.
- What is the root cause theory mentioned in figure1? Is it similar to grounded theory?
- When describing the development of lacquer art and art education in Taiwan in each section, please note the referenced documents. For example, on page 5 (not limited to this), it mentioned ‘As of 2020, there have been 68 artists who have completed 68 workshops’, but did not indicate the source of the information.
- It is mentioned in lines 404-406 that "it is called Ra-den in Chinese", but the Chinese is Luo-dian, and Ra-den is the Japanese pronunciation.
- There is also a Chinese word in line 372, please remove it.
- The way of life, the way of living, and tool of livelihood in Figure 3 are not easy to distinguish conceptually.
- Both figure4 and figure5 are important traditional craftsmen registered in the country (national treasures of the world).
- On the whole, after the revision, this article has been improved compared with the previous version, but t is recommended that the analysis of 4.1 can extract important educational concepts of lacquer craft education, and it is also recommended that the value level of 4.2 can be classified into a more clear content.
Author Response
Many thanks to the reviewer for giving us such valuable suggestions, which have enabled us to thoroughly rethink, reorganize, and propose an improved version of the study. We look forward to such amendments that will meet the reviewers' requirements. Thank you from the bottom of our hearts!
|
Q1. Although the title of the thesis mentions color education, in fact, the inheritance of lacquer art does not only teach color, but includes materials, techniques, creation and other aspects. Therefore, it is recommended that the color education in the title can be modified to craft education. A1. Thank you for your suggestion, we have finished the correction accordingly. |
|
Q2. What is the root cause theory mentioned in figure1? Is it similar to grounded theory? A2. We appreciate the reviewer's reminder that Figure 1 actually illustrates the process of this study, but the section placed in the original version seems unreasonable, so we have changed Figure 1 to the "III. Research Methodology" section in the revised version. |
|
Q3. When describing the development of lacquer art and art education in Taiwan in each section, please note the referenced documents. For example, on page 5 (not limited to this), it mentioned ‘As of 2020, there have been 68 artists who have completed 68 workshops’, but did not indicate the source of the information. Q3. Thanks to the reviewer for the reminder, in the revised version we added [22] [23] [24], etc. However, the source of the information mentioned in the study, "‘As of 2020, there have been 68 artists who have completed 68 workshops..." is internal information provided by the project staff of the Cultural Heritage Bureau, and it is not published information, so there is no indication of the source of the information here. In the revised version, we have also provided additional information. |
|
Q4. It is mentioned in lines 404-406 that "it is called Ra-den in Chinese", but the Chinese is Luo-dian, and Ra-den is the Japanese pronunciation. A4. Thank you for your suggestions, we have completed the corrections in accordance with the suggestions. |
|
Q5. There is also a Chinese word in line 372, please remove it. A5. Thanks to the reviewers for your careful review and reminders, the Chinese text has been removed in the updated version. |
|
Q6. The way of life, the way of living, and tool of livelihood in Figure 3 are not easy to distinguish conceptually. A6. We appreciate the reviewer's suggestion, and we have asked the experts who participated in this study to discuss this issue again, and we have changed it to "Tool of Livelihood", "Way of Living", and "Philosophy of Life" as the three Value Level of Craftsmanship. |
|
Q7. Both figure4 and figure5 are important traditional craftsmen registered in the country (national treasures of the world). A7. Many thanks to the reviewer for your careful review and reminder that both interviewees are indeed nationally registered and important traditional artisans, and that we have fixed this error in the revised version. |
|
Q8. On the whole, after the revision, this article has been improved compared with the previous version, but it is recommended that the analysis of 4.1 can extract important educational concepts of lacquer craft education, and it is also recommended that the value level of 4.2 can be classified into a more clear content. A8. Thank you for the valuable inputs, we have added more explanations in the updated version, which should make the overall concept clearer. |